# A Simple yet Effective $\Delta\Delta G$ Predictor is An Unsupervised Antibody Optimizer and Explainer

**Lirong Wu**[1,2], **Yunfan Liu**[1], **Haitao Lin**[1,2], **Yufei Huang**[1], **Guojiang Zhao**[2], **Zhifeng Gao**[2], **Stan Z. Li**[1,†]

[1]Westlake University, [2]DP Technology

`{wulirong, liuyunfan, linhaitao, huangyufei, stan.zq.li}@westlake.edu.cn`
`{zhaogj, gaozf}@dp.tech`

## ABSTRACT

The proteins that exist today have been optimized over billions of years of natural evolution, during which nature creates random mutations and selects them. The discovery of functionally promising mutations is challenged by the limited evolutionary accessible regions, i.e., only a small region on the fitness landscape is beneficial. There have been numerous priors used to constrain protein evolution to regions of landscapes with high-fitness variants, among which *the change in binding free energy* ($\Delta\Delta G$) of protein complexes upon mutations is one of the most commonly used priors. However, the huge mutation space poses two challenges: (1) how to improve the efficiency of $\Delta\Delta G$ prediction for fast mutation screening; and (2) how to explain mutation preferences and efficiently explore accessible evolutionary regions. To address these challenges, we propose a lightweight $\Delta\Delta G$ predictor (Light-DDG), which adopts a structure-aware Transformer as the backbone and enhances it by knowledge distilled from existing powerful but computationally heavy $\Delta\Delta G$ predictors. Additionally, we augmented, annotated, and released a large-scale dataset containing millions of mutation data for pre-training Light-DDG. We find that such a simple yet effective Light-DDG can serve as a good unsupervised antibody optimizer and explainer. For the target antibody, we propose a novel Mutation Explainer to learn mutation preferences, which accounts for the marginal benefit of each mutation per residue. To further explore accessible evolutionary regions, we conduct preference-guided antibody optimization and evaluate antibody candidates quickly using Light-DDG to identify desirable mutations. Extensive experiments have demonstrated the effectiveness of Light-DDG in terms of test generalizability, noise robustness, and inference practicality, e.g., $89.7\times$ inference acceleration and 15.45% performance gains over previous state-of-the-art baselines. A case study of SARS-CoV-2 further demonstrates the crucial role of Light-DDG for mutation explanation and antibody optimization. Codes are available in Github, and an online Platform is available for researchers.

## 1 INTRODUCRTION

Proteins usually interact with other proteins to form protein complexes that perform specific functions in biological processes (Hu et al., 2021; Wu et al., 2024b). A representative example is antibody, a $Y$-shaped protein that protects the host by binding to a specific antigen, whose binding function is mainly determined by Complementary Determining Regions (CDRs) in the antibody (Murphy & Weaver, 2016). In practice, how to mutate the amino acids on the interaction surface and select favorable mutations are two fundamental aspects of antibody optimization. There have been many antibody design methods proposed, such as MEAN (Kong et al., 2022), RefineGNN (Jin et al., 2021), and dyMEAN (Kong et al., 2023), which train conditional antibody generators on large amounts of antibody-antigen complexes and then optimize antibodies by applying *Iterative Target Augmentation* (ITA) algorithm (Yang et al., 2020b) to fine-tune the generators. Despite the great success in conditional generation for mutations, how to build an efficient evolutionary selection sieve (Hayes et al., 2024) for fast screening of mutations remains under-explored. In this paper, we shift the research focus from generating to selecting mutations and indirectly explore the underlying fitness landscape by focusing on regions where $\Delta\Delta G$s over mutations are minimized. We demonstrate that

even a simple but effective $\Delta\Delta G$ predictor can serve as a good unsupervised antibody optimizer and explainer, which doesn't require any additional functional annotations and deep generative models.

A huge challenge for protein optimization is the enormous combinatorial space of over $20^N$ potential mutations, where $N$ is the number of mutable sites. Therefore, two aspects need to be considered in the design: (1) how to develop a simple but effective $\Delta\Delta G$ predictor for fast screening of candidate mutations in a relatively short time; (2) how to explain mutation preferences and efficiently search for accessible evolutionary paths, i.e., promising mutations, from the enormous combinatorial space.

Recently, unsupervised energy-based models (Jin et al., 2023; Luo et al., 2023) have revealed that the log-likelihood of protein complexes is highly correlated with experimental binding energy, making $\Delta\Delta G$ one of the suitable priors for guiding protein evolution. Early computational approaches for $\Delta\Delta G$ prediction are mainly biophysics-based (Alford et al., 2017; Park et al., 2016; Delgado et al., 2019) or statistics-based (Geng et al., 2019; Li et al., 2016), which are limited either in efficiency or effectiveness. Recently, many deep learning-based techniques have been proposed, most of which tackle the scarcity of annotated experimental data by pre-training on massive unlabeled data using a variety of pretext tasks, including Masked Inverse Folding, Rotamer Density Estimation (RDE) (Luo et al., 2023), Side-chain Diffusion (DiffAffinity) (Yang et al., 2022), Multi-level Interaction Modeling (ProMIM) (Mo et al., 2024). Another state-of-the-art $\Delta\Delta G$ predictor is Prompt-DDG (Wu et al., 2024c), which flexibly provides wild-type and mutated complexes with their microenvironmental differences around each mutation. Despite the great advances, the architectural complexity of these methods burdens inference, which is largely due to their reliance on the IPA-style backbone as in AlphaFold2 (Jumper et al., 2021), which encodes local and global coordinates at each layer.

To develop a simple yet effective $\Delta\Delta G$ predictor (Light-DDG), it requires the fulfillment of both efficiency and effectiveness. For the goal of high inference efficiency, we simplify the architecture to a lightweight Transformer and achieve model compression by knowledge distillation. From the perspective of effectiveness, we use data augmentation techniques to compensate for the weakening of modeling capability brought by architectural simplification. To achieve this, we collected, annotated, and released a large-scale augmented dataset containing millions of mutation data for pre-training Light-DDG. A comparison of various $\Delta\Delta G$ prediction methods on the effectiveness and efficiency is presented in Fig. 1. It demonstrates the great advantages of Light-DDG, e.g., $89.7\times$

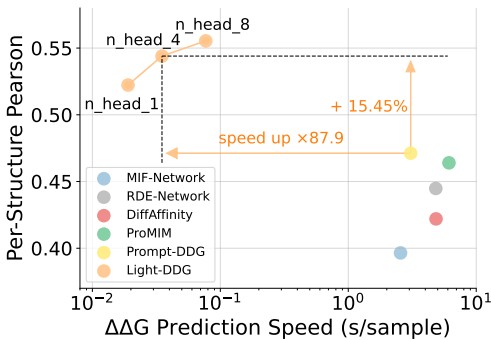

Figure 1: Efficiency vs Effectiveness. There are three variants of Light-DDG with differing numbers of attention heads (default to 4 in this paper).

inference acceleration and 15.45% performance gains over Prompt-DDG. Furthermore, we comprehensively evaluate the advantages of Light-DDG in terms of test generalizability, noise robustness, architectural applicability, and inference practicality by extensive experiments in Sec. 5.1.

Furthermore, we show that even a simple yet effective Light-DDG has the potential to be a good explainer and optimizer within a Unified framework for Antibody optimization (Uni-Anti). For the target antibody, we propose a Mutation Explainer to identify key mutation sites and learn site-wise mutation preferences. One of the design difficulties is the synergistic effect of mutations, e.g., the negative effect of a single substitution can only be tolerated in the presence of another enabling mutation (Ding et al., 2024). To tackle this problem, we develop an iterative Shapley value estimation algorithm that can measure the *marginal benefit* of each mutation per residue by coarse-to-fine iteration, while reducing the huge combinatorial space of vanilla Shapley value algorithm (Shapley et al., 1953). Based on the learned mutation preferences, we explore accessible evolutionary regions by mutation preference-guided antibody optimization and then evaluate antibody candidates quickly using Ligh-DDG. Such antibody optimization enjoys the great benefits of diversity and flexibility, capable of generating diverse antibodies with corresponding $\Delta\Delta G$ scores and rankings, and is well suited for co-optimization of multiple CDRs. Finally, we demonstrate the advantages of Uni-Anti for antibody optimization and preference explanation using a case study on SARS-CoV-2.

## 2 RELATED WORK

**Mutational Effect Prediction.** The prediction of mutation effects on single proteins has been well studied, which mainly mines co-evolutionary information from protein sequences by Multiple Sequence Alignments (MSAs) (Frazer et al., 2021; Luo et al., 2021) or Protein Language Models (PLMs) (Meier et al., 2021; Notin et al., 2022). However, predicting *the change in binding free energy* ($\Delta\Delta G$) of protein complexes upon mutations is more challenging because it involves complex interactions between proteins. Computational methods for $\Delta\Delta G$ prediction have undergone a paradigm shift from biophysics-based and statistics-based techniques (Schymkowitz et al., 2005; Park et al., 2016) to Deep Learning (DL) techniques, among which pre-training-based approaches are the most popular solutions. RDE (Luo et al., 2023) pre-trains by using a normalizing flow model to estimate the density of sidechain conformations (rotamers). Similarly, DiffAffinity (Liu et al., 2023) also models the side-chain distribution, but with a conditional diffusion model. Besides, Mo et al. (2024) proposes a multi-level pre-training framework, ProMIM, to fully capture all three levels of protein-protein interactions. Recently, Prompt-DDG (Wu et al., 2024c) proposes a microenvironment-aware hierarchical codebook that generates prompts for better $\Delta\Delta G$ prediction.

**Antibody Optimization.** Early approaches for antibody design are mostly energy-based (Adolf-Bryfogle et al., 2018; Lapidoth et al., 2015), and it is recently extended to deep generative models, including RefineGNN (Jin et al., 2021), MEAN (Kong et al., 2022), RAAD (Wu et al., 2024a), DiffAb (Luo et al., 2022), etc. These models train a conditional antibody generator and screen out a number of high-quality antibodies using a $\Delta\Delta G$ predictor. These high-quality antibodies will be used as training data to further fine-tune the antibody generator for directed antibody optimization. In this paper, we rethink the role of $\Delta\Delta G$ prediction for antibody optimization, demonstrating that a simple yet effective $\Delta\Delta G$ predictor can directly serve as a good unsupervised antibody optimizer and explainer, without requiring additional functional annotations or deep generative models.

## 3 PRELIMINARY

**Notations.** A protein complex consists of $N$ amino acid residues $(v_1, v_2, \cdots, v_N)$, where each residue $v_i$ is one of the 20 amino acid types. We characterize each residue $v_i$ with an E(3)-invariant node feature $\mathbf{x}_i = \{E_{\text{type}}(v_i), E_{\text{ang}}(v_i), E_{\text{mut}}(v_i)\}$, where $E_{\text{type}}(v_i)$ denotes the embedding of residue types, $E_{\text{angle}}(v_i)$ is the angle encoding of three dihedral angles and four torsion angles, and $E_{\text{mut}}(v_i)$ denotes the mutation embedding on whether residue $v_i$ is mutated or not. The pairwise feature between residues $v_i$ and $v_j$ is $\mathbf{e}_{i,j} = \{E_{\text{pos}}(i,j), E_{\text{dis}}(\mathbf{Z}_i, \mathbf{Z}_j), Q_i^\top \frac{\mathbf{Z}_{j,\zeta} - \mathbf{Z}_{i,C_\alpha}}{\|\mathbf{Z}_{j,\zeta} - \mathbf{Z}_{i,C_\alpha}\|} \mid \zeta\}$, where $\mathbf{Z}_i$ is the 3D coordinate of residue $v_i$, $E_{\text{pos}}(i,j)$ and $E_{\text{dis}}(\mathbf{Z}_i, \mathbf{Z}_j)$ encode the relative sequential and spatial distances between residue $v_i$ and residue $v_j$, respectively. $E_{\text{pos}}(i,j)$ is set as 0 for any two residues that are not on the same chain. Besides, the last term is the direction encoding of four backbone atoms $\zeta \in \{C_\alpha, C, N, O\}$ of residue $v_j$ in the local coordinate frame $Q_i$ of residue $v_i$ (Wu et al., 2024c). All these node and pairwise features will be pre-processed before model training.

**Transformer as the student backbone in Knowledge Distillation (KD).** To improve the inference efficiency of a $\Delta\Delta G$ predictor, a lightweight Transformer is used as the backbone to encode each protein complex $\mathcal{P} = (\mathbf{X}, \mathbf{E})$. The $l$-th ($1 \le l \le L$) layer of the Transformer is defined as follows

$$\mathbf{H}^{(l)} = \text{LN}\left(\text{FFN}\left([\text{head}_1, \cdots, \text{head}_K]\mathbf{W}_O^{(l)}\right) + \mathbf{H}^{(l-1)}\right), \text{where}$$

$$\text{head}_k = \text{softmax}\left(\frac{(\mathbf{H}^{(l-1)}\mathbf{W}_Q^{(l,k)})(\mathbf{H}^{(l-1)}\mathbf{W}_K^{(l,k)})^\top}{\sqrt{d_h}} + \mathbf{E}\mathbf{W}_E^{(l,k)}\right)\mathbf{H}^{(l-1)}\mathbf{W}_V^{(l,k)} \quad (1)$$

where $\mathbf{H}^{(0)} = \mathbf{X}$ denote the input node feature, $\mathbf{W}_O^{(l)}, \mathbf{W}_Q^{(k,l)}, \mathbf{W}_K^{(k,l)}, \mathbf{W}_V^{(k,l)}, \mathbf{W}_E^{(k,l)}$ are parameter matrices, $K$ is the number of attention heads, $\text{LN}(\cdot)$ is the layer normalization, $\text{FFN}(\cdot)$ is a two-layer feed-forward neural network with $\text{ReLu}(\cdot)$ as activation function, and $d_h$ is the hidden dimension.

**Prompt-DDG as Teacher and Annotator.** Prompt-DDG (Wu et al., 2024c) is the state-of-the-art $\Delta\Delta G$ predictor to date. During training, it trains a hierarchical prompt codebook to capture microenvironmental information at different structural scales. With the learned prompt codebook, it encodes the microenvironment around each mutation into multiple hierarchical prompts and combines them to flexibly provide information to wild-type and mutated protein complexes about their

microenvironmental differences. We use Prompt-DDG as a teacher and annotator for distillation and data augmentation in default in this paper, but also evaluate other $\Delta\Delta G$ predictors as teachers.

**Problem Statement.** Given a wild-type protein complex $\mathcal{P}_W$ and a set of mutations $\mathcal{M}$, the task of mutational effect prediction aims to learn a mapping $f(\cdot) : \mathcal{P}_W, \mathcal{M} \rightarrow \Delta\Delta G$ that encodes wild-type complex $\mathcal{P}_W$ and mutated complex $\mathcal{P}_M = g(\mathcal{P}_W, \mathcal{M})$ separately with a parameter-shared Transformer, and then feeds the difference of their pooled representations $\mathbf{h}^W$ and $\mathbf{h}^M$ into a three-layer MLP to predict the $\Delta\Delta G$ score. The objective of protein (antibody) complex optimization aims to find a mutation $\mathcal{S}$ from the mutation space $\mathbb{S}$ that minimizes $\Delta\Delta G$, that is, $\arg\min_{\mathcal{S}\in\mathbb{S}} f(\mathcal{P}_W, \mathcal{S})$.

## 4 METHODOLOGY

In this section, we propose a unified framework for directed antibody optimization with a simple but effective $\Delta\Delta G$ predictor as the core. A high-level overview of the proposed framework is shown in Fig. 3(b). We first present how to construct a large-scale augmented mutation dataset SKEMPI-Aug by cross-augmentation in Sec. 4.1. Next, we pre-train a simple but effective Light-DDG on the large-scale augmented SKEMPI-Aug dataset and then fine-tune it by knowledge distillation on the SKEMPI v2.0 dataset, as described in Sec. 4.2. Further, we propose a Mutation Explainer to learn key mutation sites and mutation preferences in Sec. 4.3, and finally introduce how to perform preference-guided mutation search in Sec. 4.4. From the perspective of the energy landscape in Fig. 3(a), Light-DDG establishes a mapping from mutations to energy changes ($\Delta\Delta G$), while Mutation Explainer iteratively explores evolutionary accessible regions based on mutation preferences.

### 4.1 A LARGE-SCALE AUGMENTED MUTATION DATASET FOR SUPERVISED PRE-TRAINING

Considering the scarcity of experimental data in the SKEMPI v2.0 dataset, pre-training on large amounts of mutations-irrelevant data has become a popular practice for training $\Delta\Delta G$ predictors. One of the most commonly used pre-training datasets is PDB-REDO (Joosten et al., 2014), in which several *unsupervised pre-training* tasks (Luo et al., 2023; Yang et al., 2022; Mo et al., 2024), have been proposed to learn generalized knowledge. However, in order to improve the inference efficiency, we use a lightweight transformer as the backbone in this paper, which has weaker modeling capability than the IPA-style backbone, making it hard to directly learn useful knowledge patterns for $\Delta\Delta G$ prediction from massive unlabeled data in an unsupervised manner. Therefore, we here consider *supervised pre-training*, but the upcoming challenge is how to construct a large-scale dataset that covers a sufficiently wide range of mutation possibilities and their corresponding $\Delta\Delta G$ scores. In this subsection, we take data augmentation as an effective means of compensating for the simplification of the ar-

Table 1: Characteristics of three datasets.

| Dataset | Size | Mutation | Pre-training |
|---|---|---|---|
| SKEMPI v2.0 | 7k | ✓ | ✗ |
| PDB-REDO | 143k | ✗ | unsupervised |
| SKEMPI-Aug | 640k | ✓ | supervised |

Figure 2: A schematic diagram of the $K$-fold cross-augmentation, where blue and red boxes indicate the separate folds used for training and data augmentation.

chitecture. To augment data, we use Prompt-DDG, the current state-of-the-art $\Delta\Delta G$ predictor, as an annotator. Specifically, we perform arbitrary mutations on several randomly selected mutation sites of complexes from the SKEMPI v2.0 dataset, feed the mutated complexes into Prompt-DDG to predict $\Delta\Delta G$ scores, and then package the mutations and predicted $\Delta\Delta Gs$ into one piece of augmentation data. To prevent data leakage, we propose $K$-fold cross-augmentation as shown in Fig. 2, where the SKEMPI v2.0 dataset is divided into $K$ equal-sized folds according to the complex structure. For each round of augmentation, we first train a new Prompt-DDG from scratch with $K$-1 folds, then augment the remaining 1 fold by random sampling and random mutation, and finally annotate it by Prompt-DDG. As a result, the data used to train Prompt-DDG is separate from the data annotated by Prompt-DDG to avoid any possible data leakage. Moreover, we set a threshold during augmentation to ensure that the augmented samples are sufficiently different from the original samples to further avoid data leakage. In such a way, we have augmented, annotated, and released a large-scale dataset called SKEMPI-Aug, which contains millions of mutation data that

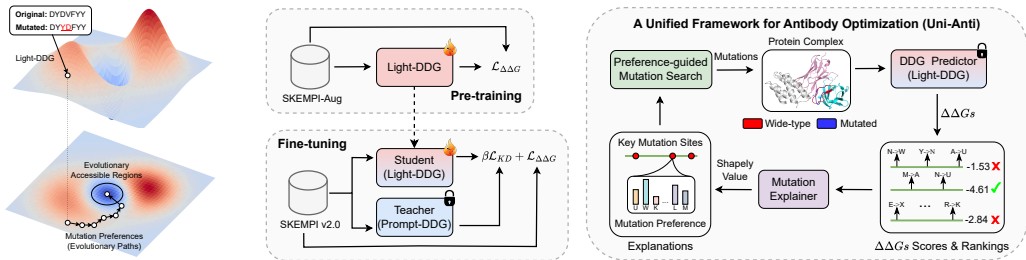

(a) Energy Landscape     (b) A High-level Overview of Uni-Anti with Light-DDG as the core.

Figure 3: (a) A binding energy landscape reflecting the mapping from mutations to $\Delta\Delta G$ scores. (b) Pre-training a Light-DDG with augmentation and distillation, and then using it as the core, together with mutation explainer and search, to construct a unified framework for antibody optimization.

can be used for supervised pre-training of $\Delta\Delta G$ predictors. We compare in Table. 1 the data sizes of three datasets, whether they include labeled mutation data, and how they are used for pre-training.

## 4.2 A Simple but Effective $\Delta\Delta G$ Predictor by Augmentation and Distillation

Knowledge Distillation (KD) is an effective means of achieving model compression (Hinton et al., 2015), and the distilled student models can even exhibit better performance than the corresponding teacher models. In this paper, we combine knowledge distillation techniques with supervised pre-training to build a simple but effective $\Delta\Delta G$ predictor (Light-DDG). Specifically, we first perform supervised pre-training on the large-scale SKEMPI-Aug dataset $\mathcal{D}_{\text{Aug}}$, and then fine-tune the model on the SKEMPI v2.0 dataset $\mathcal{D}_{\text{Skem}}$ under the joint supervision of ground-truth labels and distillation losses. Since cross-validation on SKEMPI v2.0 is performed in this paper to validate the method, the distillation objective of Light-DDG on the training data $\mathcal{D}_{\text{train}} \subseteq \mathcal{D}_{\text{Skem}}$ can be defined as:

$$f_S^* = \arg\min_{f_S'} \frac{1}{|\mathcal{D}_{\text{train}}|} \sum_{(a_i, y_i) \in \mathcal{D}_{\text{train}}} \Big( \underbrace{\left\| f_S'(a_i) - y_i \right\|^2}_{\mathcal{L}_{\Delta\Delta G}} + \beta \underbrace{\left\| f_S'(a_i) - f_T^*(a_i) \right\|^2}_{\mathcal{L}_{\text{KD}}} \Big), \quad (2)$$

where $\beta$ is a trade-off hyperparameter, $a_i = (\mathcal{P}_i, \mathcal{M}_i)$ is the input to $\Delta\Delta G$ predictor $f(\cdot)$, $y_i$ is the ground-truth $\Delta\Delta G$. In this paper, we default to Prompt-DDG as the teacher $f_T^*(\cdot)$, but we also observed significant improvements when using other $\Delta\Delta G$ predictors as teachers in the experiments. The student model $f_S'(\cdot)$ is initialized by pre-training on the SKEMPI-Aug dataset $\mathcal{D}_{\text{Aug}}$, as follows

$$f_S' = \arg\min_{f_S} \frac{1}{|\mathcal{D}_{\text{Aug}}|} \sum_{(a_i, y_i) \in \mathcal{D}_{\text{Aug}}} \underbrace{\left\| f_S(a_i) - y_i \right\|^2}_{\mathcal{L}_{\Delta\Delta G}}. \quad (3)$$

## 4.3 Mutation Explainer: Learning Mutation Sites and Mutation Preferences

A key challenge in antibody optimization is how multiple mutations combine to influence function and future mutation trajectories (Ding et al., 2024). With a simple but effective Light-DDG available, we propose a novel Mutation Explainer that can identify key mutation sites and learn mutation preferences for each residue site. This is achieved by calculating the Shapley value (Shapley et al., 1953) for each mutation at each site as its *marginal benefit*, which explains very well the *"average"* marginal contribution of each mutation across all mutation combinations. The exact Shapley value $\varphi(i, j)$ of the $i$-th ($1 \le i \le N$) site mutated to $j$-th ($1 \le j \le 20$) amino acids is defined as follows

$$\varphi(i, j) = \sum_{\mathcal{M} \subseteq \mathbb{S} \setminus (i,j)} \frac{|\mathbb{S}|!(|\mathcal{M}| - |\mathbb{S}| - 1)!}{|\mathcal{M}|!} \Big( f_S^*\big(\mathcal{P}, \mathcal{M} \cup \{(i,j)\}\big) - f_S^*\big(\mathcal{P}, \mathcal{M}\big) \Big) \quad (4)$$

The exact Shapley value $\varphi(i, j)$ is calculated by considering the effects on $\Delta\Delta G$ scores when each mutation $(i, j)$ is added or removed from the mutation set $\mathcal{M}$. However, it is impractical to enumerate all mutation possibilities in the huge mutation space $\mathbb{S}$ to calculate an exact Shapley value $\varphi(i, j)$.

For the task of antibody optimization, what we are really concerned about are those promising mutations rather than those unimportant or even harmful ones. Therefore, we propose a more efficient Iterative Shapley Value Estimation algorithm, which estimates the Shapley value of each mutation in a coarse-to-fine iterative manner, and progressively pays more attention to those promising residue sites and mutations. Specifically, the Shapley value $\widetilde{\varphi}^{(t)}(i, j)$ at $t$-th iteration is estimated as follows

$$\widetilde{\varphi}^{(t)}(i, j) = \sum_{n=1}^{D_i^{(t)}} \frac{1}{D_i^{(t)}} \Big( f_S^*\big(\mathcal{P}, \mathcal{M}_n \cup \{(i, j)\}\big) - f_S^*\big(\mathcal{P}, \mathcal{M}_n\big) \Big), \quad 1 \leq t \leq T \tag{5}$$

where $\{D_i^{(t)}\}_{i=1}^N$ are the numbers of sampling for each residue site, proportional to the site probability $p_{\text{site}}^{(t)} \in \mathbb{R}^N$. $\mathcal{M}_n$ is one mutation set that randomly selects multiple residue sites except for the $i$-th residue and mutates them according to the mutation preference $p_{\text{pre}}^{(t)} \in \mathbb{R}^{N \times 20}$. Specifically, the site probability $p_{\text{site}}^{(t+1)}$ and mutation preference for the $i$-th site $p_{\text{pre}}^{(t+1)}(i)$ is updated as follows

$$p_{\text{site}}^{(t+1)} = \alpha \cdot \sigma\Big( \sum_j \widetilde{\varphi}^{(t)}(i, j) \Big) + (1 - \alpha) \cdot p_{\text{site}}^{(t)}, \quad p_{\text{pre}}^{(t+1)}(i) = \alpha \cdot \sigma\Big( \widetilde{\varphi}^{(t)}(i, :) \Big) + (1 - \alpha) \cdot p_{\text{pre}}^{(t)}(i) \tag{6}$$

where $p_{\text{site}}^{(1)}$ and $\{p_{\text{pre}}^{(1)}(i)\}_{i=1}^N$ are initialized to be uniform distributions, $\alpha$ is the momentum updating rate, and $\sigma(\cdot) = \text{Softmax}(\cdot)$ is the activation function. Such an iterative approximation will treat every site and mutation equally at first, but gradually focus on those more potential sites and mutations to approximate the exact Shapley values as closely as possible in a limited number of samples.

### 4.4 PREFERENCE-GUIDED MUTATION SEARCH FOR ANTIBODY OPTIMIZATION

Mutation and selection are two fundamental aspects of antibody optimization. The previous popular methods usually train a deep generative model on large amounts of data, and then apply *Iterative Target Augmentation* (ITA) to guide directed optimization, i.e., generating favorable mutations. *In contrast, this paper focuses on selection rather than generation.* Given a lightweight Light-DDG and a Mutation Explainer, we can directly search for favorable mutations, requiring no additional deep generative models. For the target antibody to be optimized, we randomized 10,000 mutated antibodies by sampling mutation sites and determining mutation residues based on the site importance $p_{\text{site}}^{(T)}$ and site-wise preferences $\{p_{\text{pre}}^{(T)}(i)\}_{i=1}^N$. These mutation candidates are then quickly evaluated using a lightweight Light-DDG to get the most desirable mutations based on the rankings of their $\Delta\Delta G$ scores. We have provided pseudo-code in **Appendix A** about how Light-DDG, Mutation Explainer, and Mutation Search are constructed into a unified framework for antibody optimization.

## 5 EXPERIMENTS

**Datasets.** The effectiveness of Light-DDG for $\Delta\Delta G$ prediction is evaluated on the SKEMPI v2.0 (Jankauskaitė et al., 2019) dataset, which contains 348 complexes, 7,085 mutation combinations, and corresponding changes in binding free energy, but not any mutated complex structures. We randomly split the SKEMPI v2.0 dataset into 3 folds by complexes and perform 3-fold cross-validation for all methods. For pre-training, different pre-training-based methods use different pre-text tasks and datasets. For example, the PDB-REDO (Joosten et al., 2014) dataset contains 143k unannotated data and has been widely used for *unsupervised pre-training* by previous methods. In contrast, Light-DDG is *supervised pre-trained* on the SKEMPI-Aug datasets that consist of 670k annotated mutation data. Moreover, the AFDB dataset (Varadi et al., 2022) that contains the sequences and their corresponding structures predicted by AlphaFold2 can also be used as pre-training data.

**Evaluation Metrics.** There are seven metrics used to evaluate $\Delta\Delta G$ prediction, including *five overall metrics*: (1) Pearson correlation coefficient; (2) Spearman correlation coefficient; (3) Root Mean Squared Error (RMSE); (4) Mean Absolute Error (MAE); (5) AUROC. Additionally, (Luo et al., 2023) groups the mutations by structure, calculating the Pearson and Spearman correlation coefficients for each structure separately, and reporting the average as *two per-structure metrics*. For antibody optimization, we take the minimal $\Delta\Delta G$ score of the optimized antibodies as the metric.

**Baselines.** We compare Light-DDG with several state-of-the-art $\Delta\Delta G$ prediction methods, including ESM-1F (Hsu et al., 2022), two variants of MIF (MIF-$\Delta$logits and MIF-Network) (Yang et al.,

Table 2: Mean results of 3-fold cross-validation for $\Delta\Delta G$ prediction on the SKEMPI v2.0 dataset.

| Category | Pre-training Dataset (Szie) | Method | Per-Structure | | Overall | | | | |
|---|---|---|---|---|---|---|---|---|---|
| | | | Pearson ↑ | Spear. ↑ | Pearson ↑ | Spear. ↑ | RMSE ↓ | MAE ↓ | AUROC ↑ |
| Sequence-based | - | ESM-1v | 0.0073 | -0.0118 | 0.1921 | 0.1572 | 1.9609 | 1.3683 | 0.5414 |
| | - | PSSM | 0.0826 | 0.0822 | 0.0159 | 0.0666 | 1.9978 | 1.3895 | 0.5260 |
| | - | MSA Trans. | 0.1031 | 0.0868 | 0.1173 | 0.1313 | 1.9835 | 1.3816 | 0.5768 |
| | - | Tranception | 0.1348 | 0.1236 | 0.1141 | 0.1402 | 2.0382 | 1.3883 | 0.5885 |
| Energy Function | ✗ | Rosetta | 0.3284 | 0.2988 | 0.3113 | 0.3468 | 1.6173 | 1.1311 | 0.6562 |
| | ✗ | FoldX | 0.3789 | 0.3693 | 0.3120 | 0.4071 | 1.9080 | 1.3089 | 0.6582 |
| Supervised | ✗ | DDGPred | 0.3750 | 0.3407 | 0.6580 | 0.4687 | 1.4998 | 1.0821 | 0.6992 |
| | ✗ | End-to-End | 0.3873 | 0.3587 | 0.6373 | 0.4882 | 1.6198 | 1.1761 | 0.7172 |
| Pre-training | AFDB | ESM-1F | 0.2241 | 0.2019 | 0.3194 | 0.2806 | 1.8860 | 1.2857 | 0.5899 |
| | PDB-REDO | B-factor | 0.2042 | 0.1686 | 0.2390 | 0.2625 | 2.0411 | 1.4402 | 0.6044 |
| | | MIF-Δlogit | 0.1585 | 0.1166 | 0.2918 | 0.2192 | 1.9092 | 1.3301 | 0.5749 |
| | | MIF-Network | 0.3965 | 0.3509 | 0.6523 | 0.5134 | 1.5932 | 1.1469 | 0.7329 |
| | | RDE-Linear | 0.2903 | 0.2632 | 0.4185 | 0.3514 | 1.7832 | 1.2159 | 0.6059 |
| | | RDE-Network | 0.4448 | 0.4010 | 0.6447 | 0.5584 | 1.5799 | 1.1123 | 0.7454 |
| | | DiffAffinity | 0.4220 | 0.3970 | 0.6690 | 0.5560 | 1.5350 | 1.0930 | 0.7440 |
| | | ProMIN | 0.4640 | 0.4310 | 0.6720 | 0.5730 | 1.5160 | 1.0890 | 0.7600 |
| | SKEMPI v2.0 | Prompt-DDG | 0.4712 | 0.4257 | 0.6772 | 0.5910 | 1.5207 | 1.0770 | 0.7568 |
| Ours | SKEMPI Aug. | Light-DDG | **0.5440** | **0.5004** | **0.7429** | **0.6767** | **1.3837** | **0.9697** | **0.7935** |
| | | Δ_Prompt-DDG | +15.45% | +17.55% | +9.70% | +14.50% | +9.01% | +9.96% | +4.85% |

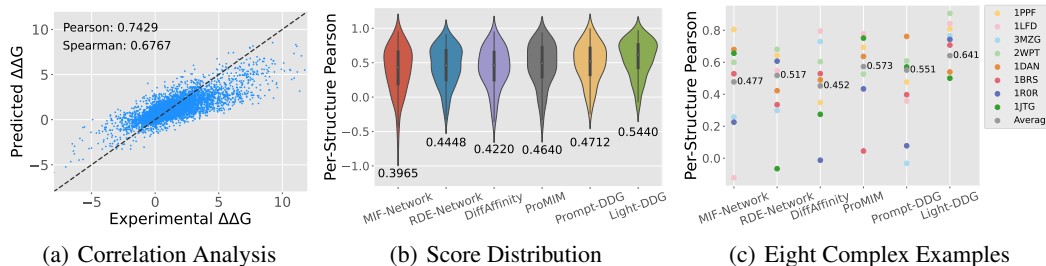

| (a) Correlation Analysis | (b) Score Distribution | (c) Eight Complex Examples |
|---|---|---|

Figure 4: (a) Correlations between experimental and predicted $\Delta\Delta G$s. (b) Distributions of per-structure Pearson scores. (c) Per-structure Pearson correlation scores for eight complex examples.

2020a), two variants of RDE (RDE-Linear and RDE-Network) (Luo et al., 2023), DiffAffinity (Liu et al., 2023), ProMIM (Mo et al., 2024), Prompt-DDG (Wu et al., 2024c), and a model that is pre-trained to predict the B-factor of residues and use predicted B-factors to predict $\Delta\Delta G$. Moreover, we compare the performance of Uni-Anti for directed antibody optimization with RefineGNN (Jin et al., 2021), MEAN (Kong et al., 2022), DiffAb (Luo et al., 2022), and dyMEAN (Kong et al., 2023). The detailed hyperparameter and implementation details can be found in **Appendix B**.

## 5.1 EVALUATION ON $\Delta\Delta G$ PREDICTION

**Performance Comparison.** Table. 2 reports 7 evaluation metrics for 18 methods on the SKEMPI v2.0 dataset, from which we observe that Light-DDG significantly outperforms all baselines on all 7 metrics, especially on the two critical per-structure metrics. For example, Light-DDG improves over Prompt-DDG on per-structure Pearson and Spearman by 15.45% and 17.55%, respectively.

**Visualizations.** The scatter plots of experimental $\Delta\Delta G$ and predicted $\Delta\Delta G$ for Light-DDG, presented in Fig. 4(a), demonstrate the strong correlation between experimental and predicted results. Besides, we provide the distribution of per-structure Pearson scores in Fig. 4(b), as well as the average results across all structures. Not only does Light-DDG achieve the best average performance, but its distribution is mostly centered on high correlation, with fewer low-correlation structures. Further, we randomly select 8 complexes and present their per-structure Pearson scores in Fig. 4(c), where Light-DDG achieves the best performance on 6 of 8 complexes.

**Single and Multiple Mutations.** We further compare Light-DDG with 7 superior methods from Table. 2 under single- and multi-point mutations, respectively. The results in Table. 3 show that two state-of-the-art methods, Prompt-DDG and ProMIM, each have strengths in different metrics and mutation settings. However, Light-DDG significantly outperforms all baselines by a large margin on 14 metrics in both mutation settings, especially more challenging multi-point mutations.

Table 3: Performance comparison of $\Delta\Delta G$ prediction under single-point and multi-point mutation.

| Method | Pre-training Dataset (Szie) | Mutations | Per-Structure | | Overall | | | | |
|---|---|---|---|---|---|---|---|---|---|
| | | | Pearson ↑ | Spear. ↑ | Pearson ↑ | Spear. ↑ | RMSE ↓ | MAE ↓ | AUROC ↑ |
| DDGPred | ✗ | single | 0.3711 | 0.3427 | 0.6515 | 0.4390 | 1.3285 | 0.9618 | 0.6858 |
| | | multiple | 0.3912 | 0.3896 | 0.5938 | 0.5150 | 2.1813 | 1.6699 | 0.7590 |
| End-to-End | ✗ | single | 0.3818 | 0.3426 | 0.6605 | 0.4594 | 1.3148 | 0.9569 | 0.7019 |
| | | multiple | 0.4178 | 0.4034 | 0.5858 | 0.4942 | 2.1971 | 1.7087 | 0.7532 |
| MIF-Network | PDB-REDO (143k) | single | 0.3952 | 0.3479 | 0.6667 | 0.4802 | 1.3052 | 0.9411 | 0.7175 |
| | | multiple | 0.3968 | 0.3789 | 0.6139 | 0.5370 | 2.1399 | 1.6422 | 0.7735 |
| RDE-Network | PDB-REDO (143k) | single | 0.4687 | 0.4333 | 0.6421 | 0.5271 | 1.3333 | 0.9392 | 0.7367 |
| | | multiple | 0.4233 | 0.3926 | 0.6288 | 0.5900 | 2.0980 | 1.5747 | 0.7749 |
| DiffAffinity | PDB-REDO (143k) | single | 0.4290 | 0.4090 | 0.6720 | 0.5230 | 1.2880 | 0.9230 | 0.7330 |
| | | multiple | 0.4140 | 0.3870 | 0.6500 | 0.6020 | 2.0510 | 1.5400 | 0.7840 |
| ProMIM | PDB-REDO (143k) | single | 0.4660 | 0.4390 | 0.6680 | 0.5340 | 1.2790 | 0.9240 | 0.7380 |
| | | multiple | 0.4580 | 0.4250 | 0.6660 | 0.6140 | 1.9630 | 1.4910 | 0.8250 |
| Prompt-DDG | SKEMPI v2.0 (7k) | single | 0.4736 | 0.4392 | 0.6596 | 0.5450 | 1.3072 | 0.9191 | 0.7355 |
| | | multiple | 0.4448 | 0.3961 | 0.6780 | 0.6433 | 1.9831 | 1.4837 | 0.8187 |
| Light-DDG | SKEMPI Aug. (670k) | single | **0.5505** | **0.5114** | **0.7328** | **0.6384** | **1.1835** | **0.8245** | **0.7777** |
| | | multiple | **0.5146** | **0.4764** | **0.7467** | **0.7343** | **1.7948** | **1.3431** | **0.8504** |

Table 4: Ablation study on knowledge distillation, data augmentation, and different input contexts.

| Method | Component | | | | Per-Structure | | Overall | | | | |
|---|---|---|---|---|---|---|---|---|---|---|---|
| | KD | Augment. | Wild Str. | Mutant Str. | Pearson ↑ | Spear. ↑ | Pearson ↑ | Spear. ↑ | RMSE ↓ | MAE ↓ | AUROC ↑ |
| Prompt-DDG | ✗ | ✗ | ✓ | ✗ | 0.4712 | 0.4257 | 0.6772 | 0.5910 | 1.5207 | 1.0770 | 0.7568 |
| Light-DDG | ✗ | ✗ | ✓ | ✗ | 0.3888 | 0.3576 | 0.6142 | 0.5244 | 1.6310 | 1.1622 | 0.7209 |
| | ✓ | ✗ | ✓ | ✗ | 0.4809 | 0.4315 | 0.7071 | 0.6297 | 1.4614 | 1.0177 | 0.7701 |
| | ✗ | ✓ | ✓ | ✗ | 0.4516 | 0.4087 | 0.6754 | 0.5796 | 1.5242 | 1.0894 | 0.7531 |
| | ✓ | ✓ | ✓ | ✗ | 0.5440 | 0.5004 | 0.7429 | 0.6767 | 1.3837 | 0.9697 | 0.7935 |
| | ✓ | ✓ | ✗ | ✗ | 0.4154 | 0.3749 | 0.6542 | 0.5590 | 1.5631 | 1.1166 | 0.7345 |
| | ✓ | ✓ | ✓ | ✓ | **0.5496** | **0.5052** | **0.7482** | **0.6824** | **1.3807** | **0.9621** | **0.7968** |

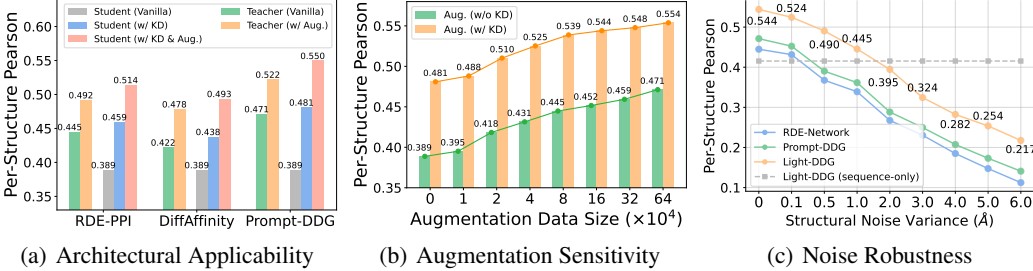

Figure 5: (a) Applicability of using different $\Delta\Delta G$ predictors as teachers. (b) Sensitivity to the sizes of augmentation data. (c) Robustness of different $\Delta\Delta G$ predictors to 3D structure Gaussian noise.

**Ablation Study.** We conduct an ablation study to evaluate the roles played by KD and augmentation. It can be observed from Table. 4 that (1) both KD and augmentation help to improve performance alone, even over Prompt-DDG (as teacher for KD and annotator for augmentation); and (2) combining the two further brings performance gains on top of each other. Furthermore, we consider two alternative input contexts, including *(i)* only wild-type and mutated sequences are available, and *(ii)* both wild-type and mutated structures are provided, where we predict mutated structures from mutated sequences by ESMFold (Lin et al., 2022). It can be found that (1) even sequence-only design performs better than energy-based and supervised baselines in Table. 2, but poorer than structure-based design, which demonstrates the importance of structural information for $\Delta\Delta G$ prediction. (2) Mutated structures only slightly improve the performance, as Prompt-DDG (teacher model) has already been implicitly pre-trained to be mutated structure-aware.

**Applicability, Sensitivity, and Robustness.** We evaluate the applicability of Light-DDG to different teachers in Fig. 5(a), where the distilled students perform better than corresponding teachers across various architectures. More importantly, it significantly improves performance regardless of whether teachers or students are pre-trained on the SKEMPI-Aug dataset. Moreover, we evaluate how the sizes of augmentation data influence Light-DDG under w/ and w/o KD settings, respec-

tively. The curves in Fig. 5(b) exhibit consistent improvements from more augmented data; however, the performance gain becomes gradually limited as the data becomes more extensive. Furthermore, we evaluate the robustness of Light-DDG to 3D structural noise by adding Gaussian noise with different variances to the wild-type structures in the inference phase. It can be found from Fig. 5(c) that the performance gets poorer with larger noise, even poorer than that of sequence-only Light-DDG. Considering that the errors of existing structure prediction are mostly around $1\mathring{A}$, in this case only structure-based Light-DDG achieves better performance than sequence-only Light-DDG.

Table 5: Rankings of the five favorable mutations on the antibody screening against SARS-CoV-2.

| Method | TH31W | AH53F | NH57L | RH103M | LH104F | Avg. Rank |
|---|---|---|---|---|---|---|
| Rosetta | 10.73% | 76.72% | 93.93% | 11.34% | 27.94% | 6.60 |
| FoldX | 13.56% | 6.88% | 5.67% | 16.60% | 66.19% | 5.80 |
| DDGPred | 68.22% | 2.63% | 12.35% | **8.30%** | 8.50% | 4.60 |
| MIF-Net. | 24.49% | 4.05% | 6.48% | 80.36% | 36.23% | 6.60 |
| RDE-Net. | 1.62% | **2.02%** | 20.65% | 61.54% | **5.47%** | 3.40 |
| DiffAffinity | 7.28% | 3.64% | 18.82% | 81.78% | 10.93% | 6.00 |
| ProMIM | 5.33% | 4.79% | 19.43% | 75.78% | 8.37% | 5.60 |
| Prompt-DDG | 2.02% | 6.88% | 3.24% | 34.81% | 6.48% | 4.00 |
| Uni-Anti (ours) | **1.21%** | 2.23% | **2.63%** | 74.90% | 6.28% | **2.40** |

Table 6: Average $\Delta\Delta G$ (kcal/mol) after antibody optimization targeted at SARS-CoV-2.

| Method | CDR-H1 | CDR-H2 | CDR-H3 | CDR-H1/2/3 | *Best* |
|---|---|---|---|---|---|
| | 7 Sites | 6 Sites | 13 Sites | 26 Sites | |
| RefineGNN | -0.473 | -1.310 | -0.086 | - | -1.310 |
| MEAN | -0.644 | -1.653 | -0.642 | - | -1.653 |
| DiffAb | -0.925 | -1.826 | -0.826 | - | -1.828 |
| dyMEAN | -0.869 | -1.942 | -0.735 | - | -1.942 |
| Random | -1.063 | -1.865 | -0.534 | -1.325 | -1.865 |
| CMA-ES | -0.972 | -1.910 | -0.683 | -1.975 | -1.975 |
| gg-dWJS | -1.124 | -1.957 | -0.770 | -2.259 | -2.259 |
| Directed | **-1.241** | **-2.192** | **-0.946** | **-2.872** | **-2.872** |

## 5.2 Antibody Screening and Optimization against SARS-CoV-2

**Candidate Antibody Screening.** The inference-efficient property of Light-DDG makes it well-suited for candidate antibody screening, i.e., identifying desirable mutations from a pool of potential mutations. We take the computational screening of 494 candidate human antibodies against SARS-CoV-2 as a case study, where all mutations are located at 26 sites within three CDRs of the heavy chain. We predict $\Delta\Delta G$s for all candidate antibodies, rank them in ascending order (lowest $\Delta\Delta G$ in the top), and report in Table. 5 the ranking of five favorable mutations that have been previously proven effective (Shan et al., 2022; Wu et al., 2024c). It can be seen that (1) Uni-Anti ranks first on two antibodies and second on the other two; (2) only Uni-Anti successfully identifies three mutations with rankings smaller than 5%; (3) Unit-Anti has the best average ranking of 2.4 among 9 methods.

**Antibody Optimization against SARS-CoV-2.** We show the effectiveness of Uni-Anti in optimizing a human anti-SARS-CoV-2 antibody to produce variants with lower binding energy. We first compare directed mutations (based on explainable mutation preferences) with random mutations in Table. 6, where we evaluate the $\Delta\Delta G$s of 10,000 sampled candidate antibodies (done with Light-DDG in less than 5 minutes) and filter out the best one. It is evident that directed mutations perform better than random mutations, especially on multi-site mutations. For example, joint random mutation of three CDRs is surprisingly inferior to mutating only CDR-H2, but directed mutation benefits remarkably from a wider range of mutation sites. Further, we compare several generative antibody optimization methods, including RefineGNN (Jin et al., 2021), MEAN (Kong et al., 2022), DiffAb (Luo et al., 2022), and dyMean (Kong et al., 2023). We input their generated antibodies together with wild-type complex structures into Light-DDG to predict $\Delta\Delta G$s. Note that these methods are conditional generative models focusing on the generation of a single CDR region, and cannot handle the joint optimization of multiple CDRs with official pre-trained models. It can be seen that regardless of which CDR region is optimized, Uni-Anti has a significant advantage over other baselines. Besides, joint optimization of three CDR regions leads to larger performance gains.

Furthe, we take Light-DDG as the fitness function and further consider two additional search strategies, gradient-guided dWJS (gg-dWJS) (Ikram et al.) and CMA-ES-based (Claussen et al., 2022). It can be observed that (1) CMA-ES-based approach has an advantage over random mutation only when the mutation space is relatively large, probably because the multivariate normal distribution in CMA-ES is not a reasonable prior for antibody mutations. (2) When optimizing a single CDR with a small mutation space, gg-dWJS slightly outperforms the current SOTA generative model, dyMEAN. However, gg-dWJS cannot benefit from such a large mutation space as Uni-Anti when jointly optimizing multiple CDRs. Last but not least, the implementation of these two approaches relies on the efficiency of Light-DDG. More results on antibody optimization can be found in **Appendix C**.

## 5.3 Visualizations on Explainable Mutation Preferences

**Single and Pairwise Mutations.** We demonstrate how Mutation Explainer can explain and guide antibody optimization, with the anti-SARS-CoV-2 antibody as an example. We present $\Delta\Delta G$s of

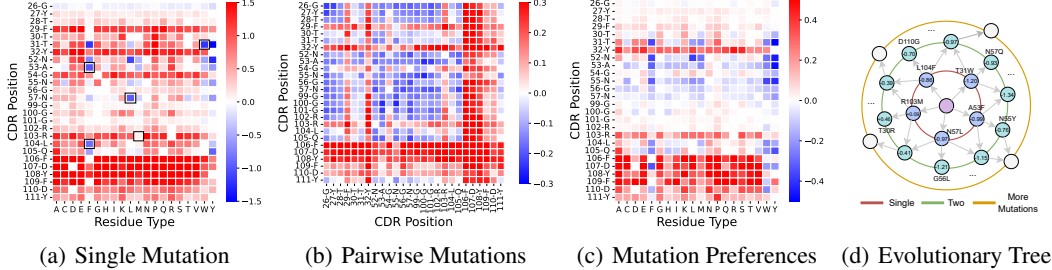

(a) Single Mutation     (b) Pairwise Mutations     (c) Mutation Preferences     (d) Evolutionary Tree

Figure 6: Visualizations on the optimization of the three CDR regions in the heavy chain for an anti-SARS-CoV-2 antibody. (a) $\Delta\Delta G$s for a single mutation. (b) Average $\Delta\Delta G$s for pairwise mutations. (c) Explainable mutation preferences based on the estimated Shapley values. (d) An example of the mutation evolutionary tree (only part of the mutations are presented for clear visualizations).

single mutation and average $\Delta\Delta G$s of paired mutations in the three CDRs of the heavy chain, respectively. It can be seen that Mutation Explainer well identifies five valid mutations (marked in black box) that have been proven effective by previous literature (Shan et al., 2022). Besides, it is clear that mutating CDR-H2 can usually lead to smaller $\Delta\Delta G$s than mutating CDR-H3 in Fig. 6(a). Moreover, pairwise mutations in Fig. 6(b) reveal important synergistic effects of mutations, i.e., a single mutation that works well may fail when occurring with other mutations (Ding et al., 2024).

**Mutation Preferences.** Considering that multiple mutations are a common application scenario, Shapley values of $\Delta\Delta G$ scores are used to estimate the marginal benefits of individual mutations, as shown in Fig. 6(c). For example, the 55-th residue on the heavy chain usually results in a negative gain when mutated alone in Fig. 6(a), but it has a small Shapley value in Fig. 6(c), suggesting its important role for multiple mutations, i.e., that it may need to work together with other mutations.

**Mutation Evolutionary Tree.** Using a lightweight $\Delta\Delta G$ predictor, along with the learned mutation preferences, we can draw a mutation evolutionary tree for the target antibody, as shown in Fig. 6(d), which is expected to provide some insights, explanations, and guidance for antibody optimization.

## 6   CONCLUSION AND DISCUSSION

This paper shifts the research focus from generating mutations to evaluating mutational effects and indirectly explores the underlying fitness landscape by focusing on regions where $\Delta\Delta G$s are minimized. To this end, we train a simple but effective $\Delta\Delta G$ predictor (Light-DDG) by data augmentation and distillation. Furthermore, we show that Light-DDG can serve as a good optimizer and explainer within a Unified framework for Antibody optimization (Uni-Anti). Extensive experiments show the superiority of Uni-Anti in mutational effect prediction, optimization, and explanation.

**Broader Impact.** The huge combinatorial space of potential mutations and the scarcity of mutation annotations have long been considered two obstacles straddling the path to unsupervised protein evolution. *On the data side*, the released augmented mutation dataset expands the original data by two orders of magnitude and is expected to be a solid data ground for follow-up works. *On the methodology side*, a lightweight $\Delta\Delta G$ predictor is expected to facilitate high-throughput fast mutation screening. In addition, the mutation preference explanations learned by Mutation Explainer can reveal the potential evolutionary paths, providing a powerful guideline for the understanding of protein functions and the discovery of high-fitness variants. Last but not least, mutation and selection are the two pillars of natural evolution. This paper provides a new perspective to achieve a novel, explainable, and unsupervised framework for directed optimization with selection at its core.

**Limitations.** Despite the great progress, several limitations still remain. Firstly, $\Delta\Delta G$ is only one common prior that constrains the evolution of proteins; combining other priors can still be built on top of our framework. Secondly, distillation is only one of the strategies to achieve lightweight inference, and other architectural choices, quantization, sparsification, and parallelization are also optional from an engineering perspective. Thirdly, the design of this paper is expected to be combined with deep generative models. We believe that (1) constructing preference pairs using Light-DDG for preference alignment and (2) taking Light-DDG as guidance in diffusion models for controllable generation are two promising solutions. Finally, more case studies on other proteins (in addition to antibodies) and wet experimental assays of the optimized antibodies will be left for future work.

## 7 ACKNOWLEDGMENTS

Many thanks to Siqi Ma for his contribution to the platform development. This work is supported by National Science and Technology Major Project (No. 2022ZD0115101), National Natural Science Foundation of China Project (No. 624B2115, No. U21A20427), Project (No. WU2022A009) from the Center of Synthetic Biology and Integrated Bioengineering of Westlake University, and Project (No. WU2023C019) from the Westlake University Industries of the Future Research Funding.

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

## A. PSEUDO CODE

The pseudo-code of the proposed Uni-Anti framework is summarized in Algorithm 1.

---

**Algorithm 1** A Unified Framework for Antibody Optimization (Uni-Anti)

---

**Input:** $M$ Wild-type Complexes and Mutations $\{(\mathcal{P}_i, \mathcal{M}_i)\}_{i=1}^M$.
1: Randomly initializing the parameters of the student model $f_S(\cdot)$.
2: # *Augmentation Pre-training*
3: Pre-training the student model $f_S(\cdot)$ on the augmented SKEMPT-Aug dataset by Eq. (3).

4: # *Training $\Delta\Delta G$ Predictor (Light-DDG)*
5: Encoding the input data with the teacher $f_T^*(\cdot)$ and the pre-trained student $f_S(\cdot)$ separately;
6: Calculating the knowledge distillation (KD) loss;
7: Fine-tuning the student $f_S(\cdot)$ by joint optimization of downstream and KD losses by Eq. (2).

8: # *Mutation Explainer*
9: Initializing $p_{\text{site}}^{(1)}$ and $\{p_{\text{pre}}^{(1)}(i)\}_{i=1}^N$ as uniform distributions.
10: **for** $t \in \{1, 2, \cdots, T\}$ **do**
11:     Calculating the shape value of each mutation at each site by Eq. (5);
12:     Updating the site importance $p_{\text{site}}^{(t+1)}$ and mutation preferences $\{p_{\text{pre}}^{(t+1)}(i)\}_{i=1}^N$ by Eq. (6).
13: **end for**

14: # *Directed Mutation Search*
15: Randomly sampling 10,000 mutation candidates based on $p_{\text{site}}^{(T)}$ and $\{p_{\text{pre}}^{(T)}(i)\}_{i=1}^N$;
16: Predicting and ranking $\Delta\Delta G$ scores of sampled mutations using the trained Light-DDG;
17: Screening out the most desirable mutations based on the rankings of their $\Delta\Delta G$ scores.

18: **return** Trained $\Delta\Delta G$ predictor (Light-DDG) and optimized antibodies.

---

## B. HYPERPARAMETERS AND IMPLEMENTATION DETAILS

Experiments are conducted based on Pytorch 1.8.0 on a hardware platform with Intel(R) Xeon(R) Gold 6240R @ 2.40GHz CPU and NVIDIA A100 GPU. The hyperparameters are as follows: learning rate $lr = 0.0003$, batch size $B = 32$, pre-training iterations $E_{\text{Aug}} = 5,000$, $\Delta\Delta G$ iterations $E_{\Delta\Delta G} = 15,000$, hidden dimension $F = 128$, number of Transformer layers $L = 4$, number of attention heads $K = 4$ (by default), loss weight $\beta = 0.1$, and momentum rate $\alpha = 0.9$. Besides, we crop sequences or structures into patches containing 32 residues by first choosing a seed residue, and then selecting its 31 nearest neighbors based on the sequential distances or the $C_\beta$-$C_\beta$ distances.

## C. ANTIBODY OPTIMIZATION ON SABDAB

To further demonstrate Uni-Anti's effectiveness in antibody optimization in addition to anti-SARS-CoV-2 antibody, we further optimize CDR-H3 of 500 antigen-antibody complexes from the SAbDab dataset (Dunbar et al., 2014) and report the average (optimal) $\Delta\Delta G$ scores of various baselines in Table. A1. For RefineGNN, MEAN, dyMEAN, we incorporate Iterative Target Augmentation (ITA) (Yang et al., 2020b) into the optimization process to fine-tune the generators. For DiffAb, we directly generated 10,000 candidate samples and then selected the best one. For all baselines, we feed their optimized sequence together with wild-type complex structures into Light-DDG to predict $\Delta\Delta G$s. The results in Table. A1 show that Uni-Anti achieves superior results with a notably lower average $\Delta\Delta G$ score, demonstrating its potential advantages in terms of antibody optimization.

Table A1: Average $\Delta\Delta G$s after optimizing CDR-H3 of 500 antibodies from the SAbDab dataset.

| Method | Refine-GNN | MEAN | DiffAb | dyMEAN | Uni-Anti (ours) |
|---|---|---|---|---|---|
| $\Delta\Delta G\downarrow$ | -2.16 | -2.73 | -2.84 | -3.05 | -3.87 |

