# OpenReview forum: "A Simple yet Effective $\Delta\Delta G$ Predictor is An Unsupervised Antibody Optimizer and Explainer"
_ICLR.cc/2025/Conference — ICLR 2025 Poster_

### Official Review · Reviewer_Qvcz · 2024-10-30

**Soundness:** 3
**Presentation:** 4
**Contribution:** 3
**Rating:** 6
**Confidence:** 3

**Summary:**

This paper proposes a simple and effective Light-DDG model to predict the binding free energy change of protein complexes. The method utilizes an augmented large-scale dataset as supervised pretraining and achieves SoTA performance through a simple knowledge distillation strategy. Additionally, a novel mutation explainer is proposed to design Uni-Anti, an antibody optimization framework that enables fast screening with Light-DDG, and the effectiveness of the framework is demonstrated through a case study of SARS-CoV-2.

**Strengths:**

1. This paper demonstrates the high predictive performance of binding free energy changes using a lightweight transformer-based predictor, enabled by supervised pretraining on a large-scale mutation dataset and knowledge distillation.
2. The authors introduced an iterative Shapley value estimation algorithm, a coarse-to-fine iterative approach that effectively identifies key mutation sites, replacing the traditional Shapley value algorithm.
3. The Uni-Anti framework enables rapid antibody optimization through fast screening without requiring a generative model, offering a practical solution for antibody design.

**Weaknesses:**

1. The backbone lightweight Transformer processes E(3)-invariant features, but the paper does not discuss whether the model guarantees E(3)-invariant outputs.
2. Although the iterative Shapley value estimation algorithm is proposed to efficiently explore a large mutation space, it may lack a mathematical guarantee of accurately approximating Shapley values.

**Questions:**

1.	Could you clarify how the mutated structures (Mutant Str.) of Light-DDG in Table 4 are provided?
2.	While Prompt-DDG is a SoTA model, there may be potential for inaccuracies in pseudo-labeling. If so, would any methods be considered to address these?
3.	Additionally, since augmentation could produce samples highly similar to SKEMPI v2.0, is there a filtering step in place to prevent overlaps? If not, would this be a consideration?
4.	The dWJS [1] might apply to the antibody optimization task. Would a comparative analysis be possible?
5.	The reference to the SAbDab dataset on p9 may be incorrect. Could you please verify?

[1] Frey et al. (2024) "Protein Discovery with Discrete Walk-Jump Sampling", ICLR

---

> ### Author Response · Authors · 2024-11-18
> **Response to Reviewer Qvcz**
>
> Thanks for your insightful comments! We have marked the revised or newly added contents in red in the revised manuscript, and address the concerns you raised one by one as below:
>
> ---
>
> **Q1**: The paper does not discuss whether the model guarantees E(3)-invariant outputs.
>
> **A1**: E(3)-invariant denotes invariance to 3D translational and rotational transformations. Given that the Transformer model used in this paper does not involve explicit manipulation of spatial coordinates, inputting E(3)-invariant features can guarantee E(3)-invariant outputs.
>
> ---
>
> **Q2**: It may lack a mathematical guarantee of accurately approximating Shapley values.
>
> **A2**: We sincerely appreciate the reviewer's valuable comment. We acknowledge that the lack of a mathematical guarantee is a weakness. However, we hope the reviewer can understand that in dealing with such a large and complex mutation space problem, it is extremely challenging to provide strict mathematical proof for the accurate approximation of Shapley values.
>
> Our design is more heuristic-driven. As stated in the paper, "what we are really concerned about are those promising mutations rather than those harmful ones.” In other words, our algorithm is heuristically designed to gradually focus on more promising mutations to obtain a reasonable approximation of Shapley values for them within limited computational resources and time rather than precisely estimating Shapley values of all mutation possibilities. Although a strict mathematical proof is difficult, we have extensive empirical results to support the algorithm's effectiveness. For example, the algorithm can learn reasonable mutation preferences and identify five effective mutations (See **Figure 6(a)** marked by black boxes). Moreover, in the case of antibody optimization against SARS-CoV-2, directed mutations guided by the iterative Shapley value estimation algorithm showed great advantages over random mutations (See **Table 6**), especially with multi-site mutations. In our opinion, we hope these empirical results compensate to some extent for the lack of a mathematical proof. In the future, we’d like to explore more suitable mathematical theories to support our algorithm or further improve the algorithm to make it closer to the exact calculation of Shapley values.
>
> ---
>
> **Q3**: Could you clarify how the mutated structures (Mutant Str.) of Light-DDG in Table 4 are provided?
>
> **A3**: Since we lack access to the ground-truth mutated structures, we predict mutated structures from mutated sequences by ESMFold, as described in **Lines 422-423** of the manuscript.
>
> ---
>
> **Q4**: While Prompt-DDG is a SoTA model, there may be potential for inaccuracies in pseudo-labeling.
>
> **A4**: Firstly, we think that completely eliminating incorrect pseudo-labels is not feasible in the case where we do not have access to the ground-truth annotations. Secondly, those slightly incorrect pseudo-labels sometimes play an important role during knowledge distillation, i.e., “dark knowledge” as described in [1]. However, those severely mislabeled pseudo-labels are indeed detrimental to the model training, so a filtering process (e.g., based on the extent of deviation from the original) can be applied to remove those obvious pseudo-labeled “outliers”.
>
> [1] Hinton G. Distilling the Knowledge in a Neural Network[J]. 2015.
>
> ---
>
> **Q5**: Since augmentation could produce samples highly similar to SKEMPI v2.0, is there a filtering step in place to prevent overlaps?
>
> **A5**: In practice, we set a threshold to control the similarity between the augmented and original samples so as to avoid excessive overlap. In other words, only the augmented samples whose mutations are sufficiently different from the original will be added to the augmented dataset. The above explanations have been added in **Line 251** of the revised manuscript.
>
> ---
>
> **Q6**: The dWJS might apply to the antibody optimization task. Would a comparative analysis be possible?
>
> **A6**: Ikram et al. propose a gradient-based dWJS (gg-dWJS) that can be used for antibody sequence optimization [1]. gg-dWJS initially focuses on beta sheets and protein instability, so we adapted it for the task in **Table 6** by using Light-DDG as a discriminator. The new comparison results have been added to **Table 6**. We observe that gg-dWJS achieves slightly better performance than the current SOTA generative model, dyMEAN, for the optimization of single CDR (small mutation space). However, while gg-dWJS can be used for joint optimization of multiple CDRs, it cannot benefit from such a large mutation space as Uni-Anti did.
>
> [1] Ikram, et, al. Antibody sequence optimization with gradient-guided discrete walk-jump sampling. In ICLR 2024 Workshop.
>
> ---
>
> **Q7**: The reference to the SAbDab dataset on p9 may be incorrect. Could you please verify?
>
> **A7**: The reference to the SAbDab dataset has been corrected.
>
> ---
>
> We sincerely hope that you can appreciate our efforts on responses and revisions.

---

> ### Author Response · Authors · 2024-11-25
>
> Dear Reviewer Qvcz,
>
> Thank you for your insightful and helpful comments once again. We have thoroughly considered all the feedback and provided a point-to-point response to you.
>
> The deadline for author-reviewer discussions is approaching. If you still need any clarification or have any other questions, please do not hesitate to let us know. We hope that our response addresses your concerns to your satisfaction and we would like to kindly request a reconsideration of a higher rating if it is possible.
>
> Best regards,
>
> Authors.

---

> > ### Comment · Reviewer_Qvcz · 2024-11-25
> > **Official Comment by Reviewer**
> >
> > I have thoroughly reviewed the revisions provided by the authors, as well as the feedback from other reviewers. While I have decided to maintain my score, I do not feel strongly about the acceptance of this submission due to its lack of mathematical rigor and reliance on heuristic approximations.

---

> > > ### Author Response · Authors · 2024-12-02
> > >
> > > We sincerely appreciate your valuable feedback. We recognize that the heuristic approximation is indeed not perfect, but it is an interesting attempt to take the next step. In the future, we hope to explore more suitable mathematical theories to support our algorithm or to further improve it.
> > >
> > > If you still have questions about this, please do not hesitate to contact us.

---

### Official Review · Reviewer_tfwL · 2024-10-30

**Soundness:** 4
**Presentation:** 3
**Contribution:** 4
**Rating:** 8
**Confidence:** 3

**Summary:**

This paper introduces a transformer architecture to predict $\Delta\Delta G$ (the differences in free energy binding due to mutation).  They use knowledge distillation to rather surprisingly obtain a much small, yet more accurate model than the large teach model it is trained on.  They demonstrate that having a light-weight $\Delta\Delta G$ predictor allows them to solve some important downstream tasks involving protein mutation prediction, protein evolution, etc.  There results beat the current state-of-the-art by a significant margin.

**Strengths:**

This is a strong paper with impressive results.  There should be of interest to people working in learning representation as it shows the effectiveness of using a transformer model to extract knowledge from a much larger model.  The experiments conducted are comprehensive and convincing.  There is a nice ablation study and various real world case studies showing the utility of the predictor.

**Weaknesses:**

I found the title difficult to comprehend because of the use of $\Delta \Delta G$, which I had no idea about.  Maybe this is common place and just reflects my ignorance, but it slowed down my ability to understand the paper.  This is a very minor comment which I'm happy for the authors to ignore.

**Questions:**

Why does Light-DDG do so much better than Prompt-DGG given that it is trained using Prompt-DGG?  I know that there are claims that pupils in knowledge-distillation can be better than the teacher, but my experience is that this is rare and for there to be such a large improvement is unusual.  I am assuming that Light-DDG include components not in Prompt-DGG, but I did not pick up what makes the big difference.

I'm also a bit lost why predicting $\Delta \Delta G$ should have such an impact.  Is the binding free energy the overwhelming factor in determining the effectiveness of an antibody?

My intuition would be that predicting $\Delta \Delta G$ would be very difficult as any mutation could cause a significant shape change which is hard to predict.  Thus, I would have thought your predictor would lead to only a moderate improvement.  Clearly my intuition or logic is wrong.  Where am I wrong (or why does your model work so well)?

---

> ### Author Response · Authors · 2024-11-18
> **Response to Reviewer tfwL**
>
> Thanks for your insightful comments! We have marked the revised or newly added contents in red in the revised manuscript, and address the concerns you raised one by one as below:
>
> ---
>
> **Q1**: I'm also a bit lost why predicting ΔΔG should have such an impact. Is the binding free energy the overwhelming factor in determining the effectiveness of an antibody?
>
> **A1**: The most important property of an antibody, compared to a protein in general, is its specificity, i.e., its capability to bind strongly to the target antigen but not others. One of the most common priors used to measure the binding strength is binding energy, making ΔΔG (the change in binding free energy upon mutations) a useful practice for antibody optimization.
>
> Many previous works on antibody design [1,2,3,4] have also adopted ΔΔG to determine the effectiveness of antibody optimization. Finally, as we explain in Limitations, "ΔΔG is only one common prior that constrains the evolution of proteins; combining other priors can still be built on top of our framework."
>
> [1] Kong, et. al. Conditional antibody design as 3d equivariant graph translation. ICLR 2023.
>
> [2] Kong, et. al. End-to-end full-atom antibody design. ICML 2023.
>
> [3] Tan, et. al. Cross-Gate MLP with Protein Complex Invariant Embedding is A One-Shot Antibody Designer. AAAI 2024.
>
> [4] Lin. et. al. Geoab: Towards realistic antibody design and affinity maturation. ICML 2024.
>
> ---
>
> **Q2**: Why does Light-DDG do so much better than Prompt-DGG given that it is trained using Prompt-DGG? Why does your model work so well?
>
> **A2**: What makes Light-DDG outperform Prompt-DDG by a wide margin is twofold:
>
> - (1) more augmentation data enables the model to explore a larger mutation space more fine-grainedly, and “see” more combinations of mutations;
> - (2) knowledge distillation, which enables the model to learn meaningful knowledge (esp. structure-aware knowledge) from Promopt-DDG.
>
> When we consider only the role of knowledge distillation, as evidenced by the ablation study presented in **Table 4**, it helps but only enables the student to **slightly** outperform its teacher (Prompt-DDG), e.g., from 0.4257 to 0.4315 in per-structure Spearman. Similarly, if the model merely learns general mutation patterns from augmented data without guidance from Prompt-DDG by distillation, the resulted model only achieves comparable performance to Prompt-DDG.
>
> However, when we integrate both of these factors, a huge performance improvement can be achieved. We attribute this to the distinct roles played by augmentation and distillation in the whole process. Augmentation enables the model to “see” a wider range of mutation combinations, thereby enhancing its ability to learn general mutation patterns and improving its generalizability. Subsequently, given the distributional shift between augmented and original samples, the information from Prompt-DDG serves as a regularization for distributional alignment, facilitating a better transfer of learned general patterns to the test data.
>
> ---
>
> In light of these responses, we hope we have addressed your concerns. If we have left any notable points of concern unaddressed, please do share and we will attend to these points. We sincerely hope that you can appreciate our efforts on responses and revisions and thus raise your score.

---

> ### Author Response · Authors · 2024-11-25
>
> Dear Reviewer tfwL,
>
> Thank you for your insightful and helpful comments once again. We have thoroughly considered all the feedback and provided a point-to-point response to you.
>
> The deadline for author-reviewer discussions is approaching. If you still need any clarification or have any other questions, please do not hesitate to let us know.
>
> Best regards,
>
> Authors.

---

> > ### Comment · Reviewer_tfwL · 2024-11-25
> >
> > I have read your response.  Thank you for your clarifications.  I felt and still feel that this is a strong paper that is worthy of publication.  I think the score of 8 is still appropriate.

---

> > > ### Author Response · Authors · 2024-12-02
> > >
> > > We sincerely appreciate your valuable comments. It is encouraging to hear from you that “this is a strong paper that is worthy of publication”.

---

### Official Review · Reviewer_Ceec · 2024-11-03

**Soundness:** 2
**Presentation:** 2
**Contribution:** 2
**Rating:** 5
**Confidence:** 4

**Summary:**

This paper introduces Light-DDG, a lightweight ΔΔG predictor for protein-protein interactions, with a focus on antibody optimization. The authors address the challenge of efficient mutation screening by developing a simpler architecture that achieves faster inference while maintaining predictive performance through two key techniques: (1) knowledge distillation from more complex models like Prompt-DDG, and (2) supervised pre-training on an augmented dataset (SKEMPI-Aug) they create and release.
The main contributions include:

- A lightweight Transformer-based ΔΔG predictor that achieves 89.7× faster inference compared to state-of-the-art models
- A large-scale augmented mutation dataset (SKEMPI-Aug) containing ~640k annotated samples for supervised pre-training
- An approach for explaining mutation preferences using iterative Shapley value estimation
- *in silico* Experimental validation showing competitive or improved performance across multiple metrics, including per-structure Pearson correlation and RMSE
- A case study demonstrating the model's application to SARS-CoV-2 antibody optimization

The authors demonstrate their method's effectiveness through *in silico* experiments, including ablation studies examining the impact of knowledge distillation and data augmentation, as well as analyses of structural noise robustness and architectural applicability.

**Strengths:**

1. Demonstrates effective application of augmentation and knowledge distillation to achieve significant inference speedup (89.7x) while maintaining or improving accuracy
2. The simplified architecture enables interpretable explainability results (Figure 6)
3. Shows promising results for combining architectural simplification with performance improvements through principled use of ML techniques

**Weaknesses:**

1. **Dataset and Baseline Comparison Issues**
   The paper's central claims about performance improvements are undermined by inconsistent dataset usage across baselines. The ablation studies clearly demonstrate the outsized impact of the dataset choice, yet there is no clear indication that all baseline models were trained on equivalent data. This makes the comparative results in Table 2 difficult to interpret meaningfully.

2. **Data Leakage Concerns**
   There are serious methodological concerns about potential data leakage in the experimental setup. The authors use Prompt-DDG both as a teacher model and for dataset augmentation, but Prompt-DDG was itself trained on SKEMPIv2.0. This creates a circular dependency where test set information may be leaking into the training process through the augmented data, regardless of splitting schemes. The authors claim to avoid this via the "cross-augmentation" scheme [a technique I'm not aware of, nor was able to find reference for), depicted in Figure 2. However, I believe the authors need to explicitly address how they prevented this contamination from occurring, particularly given the ablation results that show the outsized impact of the augmentation on performance.

3. **Inadequate Baseline Comparisons for Mutation Search**
   The authors' approach of screening 10,000 random candidates significantly understates the complexity of the protein design problem and ignores established optimization algorithms. Given their efficient ΔΔG predictor as a fitness function, they should have compared against approaches like:
   • CMA-ES (Covariance Matrix Adaptation Evolution Strategy)
   • Gradient-guided Gibbs sampling
   • Basic evolutionary algorithms with standard mutation operators

The strong performance of their random baseline in Table 6 suggests these evolutionary approaches could be quite effective.

4. **Overstatement of Framework**
   The authors' claim of developing a "unified framework" for antibody optimization overstates their contribution. What they demonstrate is primarily a successful application of knowledge distillation and data augmentation to improve model efficiency and generalization. While valuable, this falls short of the comprehensive framework they suggest.

**Questions:**

1. Can you provide evidence that all baseline models were trained on equivalent datasets to ensure fair comparison?
2. How specifically did you prevent data leakage given Prompt-DDG's training history on SKEMPIv2.0?
3. Can you provide comparisons against established optimization methods like CMA-ES and gradient-guided sampling?
4. What justifies presenting this as a "unified framework" rather than an efficient implementation of a ΔΔG predictor?

---

> ### Author Response · Authors · 2024-11-18
> **Response to Reviewer Ceec - Part (1/2)**
>
> Thanks for your insightful comments! We have marked the revised or newly added contents in red in the revised manuscript, and address the concerns you raised one by one as below:
>
> ---
>
> **Q1**: Can you provide evidence that all baseline models were trained on equivalent datasets to ensure fair comparison?
>
> **A1**: Two types of datasets are involved in **Table 2**, which are (1) (optional) pre-training dataset; and (2) cross-validation dataset. For cross-validation, **Table 2** ensures a fair comparison by reporting mean results of three-fold cross-validation on the same dataset, SKEMPI v2.0, for all baselines.
>
> For pre-training, not all approaches require pre-training data. Even among pre-training-based methods, it is not feasible for all methods to use equivalent data for pre-training, as **the choice of pre-training data depends on the design of the pre-training task**. For example, MIF, RDE, DiffAffinity, and ProMIM propose new unsupervised pre-training tasks, thus using the PDB-REDO dataset for pre-training. In addition, ESM-1F takes inverse folding as a pre-training task, so the AFDB dataset is used as the pre-training data. Prompt-DDG, on the other hand, aims to learn a prompt codebook directly on SKEMPI V2.0 in an unsupervised manner. In contrast, this paper doesn't propose any new unsupervised pre-training task, but focuses on constructing a new dataset by data augmentation for supervised pre-training with augmented annotations.
>
> To make this point clearer, we have added the relevant pre-training data in **Table 2**. Also, we describe in detail the three pre-training datasets involved in **Table 2** in **Lines 311-316**.
>
> ---
>
> **Q2**: How specifically did you prevent data leakage given Prompt-DDG's training history on SKEMPIv2.0?
>
> **A2**: Data leakage is a matter of significant concern to us. Indeed, we have dedicated a considerable amount of space to discussing it within the submitted manuscript. However, based on your comments, we realized it wasn't entirely clear, so we've added more details to the revised manuscript. Further, we would like to address your concern from the following two aspects.
>
> - **How we prevent data leakage during augmentation?** To achieve this, we propose a novel cross-augmentation strategy. You have not found the reference because it is original to this paper, and it is named “cross-augmentation” due to its similarity to cross-validation. Firstly, we divide the entire SKEMPI v2.0 into K folds to ensure that there is no overlap of proteins between any two folds. Given the training history of Promot-DDG on SKEMPI v2.0, we did not directly use its original pre-trained model as an annotator, but instead performed K rounds of cross-augmentation. For each round of augmentation, we train a new Prompt-DDG from scratch with K-1 folds, and then augment the remaining 1 fold and annotate it. This way, the data used for training Prompt-DDG and the data annotated by Prompt-DDG are separated, avoiding data leakage. Moreover, we set a threshold during augmentation to ensure that the augmented samples are different from the original ones, which further avoids data leakage. The above explanations have been added in **Lines 211-215** of the revised manuscript.
>
> - **How we prevent data leakage during distillation?** We fairly compare various methods in **Table 2** by cross-validation on SKEMPI v2.0. In each round of validation, we train a new Prompt-DDG from scratch with K-1 folds, then perform distillation also on these K-1 folds, and finally peform testing on the remaining 1 fold. This way, the data used for training Prompt-DDG and distillation and the data for testing are separated, avoiding data leakage. We have revised the description of **Eq. (2)** to highlight cross-validation and make it clear.

---

> ### Author Response · Authors · 2024-11-18
> **Response to Reviewer Ceec - Part (2/2)**
>
> **Q3**: Can you provide comparisons against established optimization methods like CMA-ES and gradient-guided sampling?
>
> **A3**: Thank you for your valuable comments. To address the issue of inadequate baseline comparisons, we have added the following two additional experiments:
>
> **Experiment (1):**  We provide a performance comparison of random search and (preference-based) directed search under different numbers of mutation candidates. The results are shown below. It can be observed that random search nearly fails with limited searches, whereas preference-based search performs quite well, which demonstrates its advantage of searching for promising mutations with limited resources. Besides, random search can benefit from more searches, e.g., ΔΔG from -1.865 to -2.316 when the mutation candidates are increased from 10,000 to 10,0000. However, it should be noted that the prerequisite that random search works well in the huge mutation space is the fast screening enabled by Light-DDG of this paper.
>
> | #Candidate | Method | CDR-H1 | CDR-H2 | CDR-H3 | CDR-H1/2/3 |
> | --- | ---| --- | --- | --- | --- |
> | 100        | Random | -0.173 | -0.579 | -0.042 | -0.215 |
> | 100        | Directed | -0.864 | -1.631 | -0.653 | -2.081 |
> | 1,000     | Random | -0.643 | -1.422 | -0.195 | -0.832 |
> | 1,000     | Directed | -1.179 | -2.036 | -0.818 | -2.419 |
> | 10,000   | Random | -1.063 | -1.865 | -0.534 | -1.325 |
> | 10,000   | Directed | -1.241 | -2.192 | -0.946 | -2.872 |
> | 100,000 | Random | -1.320 | -2.316 | -0.730 | -1.418 |
> | 100,000 | Directed | -1.418 | -2.408 | -1.106 | -3.151 |
>
> **Experiment (2):**  Taking Light-DDG as the fitness function, we further compare two additional search strategies, gradient-guided sampling and CMA-ES-based, in **Table 6**. For the former, we directly compare the Gradient-guided Discrete Walk-Jump Sampling (gg-dWJS) proposed in [1]. For the latter, we did not find a direct application of CMA-ES to antibody, so we refer to the implementation of [2] and transfer it from peptide complexes to antigen-antibody complexes. It can be observed that (1) CMA-ES-based approach has an advantage over random mutation only when the mutation space is relatively large, probably because the multivariate normal distribution in CMA-ES is not a reasonable prior for the ground-truth antibody evolution path. (2) We observe that gg-dWJS achieves slightly better performance than the current SOTA generative model, dyMEAN, for the optimization of single CDR (small mutation space). However, while gg-dWJS can be used for joint optimization of multiple CDRs, it cannot benefit from such a large mutation space as Uni-Anti did. Last but not least, the implementation of these two approaches relies on the efficiency of Light-DDG. This experiment has been added to **Table 6**, and the analysis is in **Lines 474-482**.
>
> | Method | CDR-H1 | CDR-H2 | CDR-H3 | CDR-H1/2/3 |
> | ---| --- | --- | --- | --- |
> | RefineGNN | -0.473 | -1.310 | -0.086 | - |
> | MEAN | -0.644 | -1.653 | -0.642 | - |
> | DiffAb | -0.925 | -1.826 | -0.826 | - |
> | dyMEAN | -0.869 | -1.942 | -0.735 | - |
> | Random  | -1.063 | -1.865 | -0.534 | -1.325 |
> | CMA-ES  | -0.972 | -1.910 | -0.683 | -1.975 |
> | gg-dWJS | -1.124 | -1.957 | -0.770 | -2.259 |
> | Directed  | **-1.241** | **-2.192** | **-0.946** | **-2.872** |
> | ΔΔ$_{gg-dWJS}$ | -10.4% | -12.0% | -22.9% | -27.1% |
>
> [1] Ikram, et, al. Antibody sequence optimization with gradient-guided discrete walk-jump sampling. In ICLR 2024 Workshop.
>
> [2] Claussen, et al. CMA-ES-Rosetta: Blackbox optimization algorithm traverses rugged peptide docking energy landscapes. bioRxiv, 2022.
>
> ---
>
> **Q4**: What justifies presenting this as a "unified framework" rather than an efficient implementation of a ΔΔG predictor?
>
> **A4**: The efficient implementation of the ΔΔG predictor is one of the main contributions of this paper, yet it is beyond that. A lightweight ΔΔG predictor makes two sub-tasks for antibody optimization possible: (1) mutation preference learning and (2) mutation search. For the former (1), ΔΔG predictor is a crucial component of Eq. (5) for calculating Shapley values. For the latter (2), it relies on a ΔΔG predictor to rank candidate mutations to determine the “final” antibodies. That is, with an efficient ΔΔG predictor at the core, the three tasks related to antibody optimization, evaluation (ΔΔG prediction), explanation (mutation preferences), and mutation (mutation search), are unified in a single framework, as illustrated in **Figure 3(b)**.
>
> To summarize, what we mean by “unified framework” here is that one framework is used for three tasks, all of which rely on a sufficiently efficient ΔΔG predictor. We hope the above explanation can alleviate your concerns. If you insist that “unified” cannot be used, we would be willing to revise the expression in the manuscript.
>
> ---
>
> We sincerely hope that you can appreciate our efforts on responses and revisions. If you still have any further concerns, please feel free to contact us.

---

> ### Author Response · Authors · 2024-11-25
>
> Dear Reviewer Ceec,
>
> Thank you for your insightful and helpful comments once again. We have thoroughly considered all the feedback and provided a point-to-point response to you.
>
> The deadline for author-reviewer discussions is approaching. If you still need any clarification or have any other questions, please do not hesitate to let us know. We hope that our response addresses your concerns to your satisfaction and we would like to kindly request a reconsideration of a higher rating if it is possible.
>
> Best regards,
>
> Authors.

---

> > ### Comment · Reviewer_Ceec · 2024-11-25
> > **Apologies for late reply**
> >
> > Dear authors,
> >
> > Thank you for your thorough responses addressing the baselines and optimization comparisons in your manuscript. I also want to acknowledge several positives in your revisions:
> >
> > - The expanded optimization comparisons with CMA-ES are valuable.
> > - Your clarification about pre-training vs. cross-validation datasets helps frame the baseline comparisons better.
> > - The directed vs. random search analysis across different candidate pools is informative.
> >
> > However, there remains the issue of:
> > **Data Leakage in Cross-Augmentation Strategy**
> >
> > Your proposed "cross-augmentation" strategy introduces a significant data leakage problem that persists despite fold splitting. Unlike traditional cross-validation, which strictly separates training and testing data to accurately estimate generalization error, your approach generates new training data using model predictions. Even though separate models are trained for each fold, they share the same architecture and training objectives, leading to the learning of similar patterns regarding mutation effects. Consequently, the augmented data inherently carry the models' biases and assumptions about mutation effects, allowing information from the test set to inadvertently influence the training process. This undermines the validity of your results, as the apparent performance improvements may stem from this contamination rather than true generalization.
> >
> > Furthermore, comparing your approach to large-scale augmentation methods like AlphaFold 3 is not reasonable. AlphaFold 3 benefits from extensive data augmentation on a massive scale, which is fundamentally different from your method. The scale and nature of AlphaFold's data augmentation provide robustness that your current strategy does not achieve, making such a comparison inappropriate.
> >
> >
> > Your revisions have strengthened several aspects of your manuscript, and the overall method is promising, so I've raised my score to a 5. However, the unresolved data leakage issue critically impacts the robustness and reliability of the proposed method. As this problem is essential for the validity of your findings and to ensure that your results reflect genuine predictive performance, I cannot raise my score further
> >
> > While I'm not saying I'm certain the augmentation method doesn't work, based on what you've shown, this entirely novel statistical method that forms the backbone of your method's performance (as far as I can tell from the ablations) requires more validation at a fundamental level.

---

> ### Author Response · Authors · 2024-11-27
>
> Dear Reviewer Ceec,
>
> Thanks for your prompt response. We are delighted to know that you appreciate the revisions we made to the manuscript. Concerning the few issues that still concern you, we would like to provide further clarification. Firstly, we elaborate on the measures we have taken to avoid data leakage in the cross-augmentation. Subsequently, we provide detailed, point-by-point responses to address your concerns. Finally, we append an extra experiment and present the corresponding analysis.
>
> **What we have taken to avoid data leakage in cross-augmentation from two levels of data.**
>
> Any one piece of augmentation data consists of two components: mutation and label (ΔΔG).
> - **There is no data leakage in mutation patterns**. *The ground-truth mutations in that fold to be augmented have not been directly predicted and used as augmentation data.* As depicted in Figure 2, the "S2: augmentation" (involving random sampling and random mutation) was carried out prior to "S3: prediction", i.e., "we perform arbitrary mutations on several randomly selected mutation sites". Consequently, the mutation patterns underlying the augmented data are hand-crafted and random, and fundamentally different from the ground-truth mutation patterns in the training data that adhere to the natural distribution. Additionally, we set a threshold during augmentation to ensure that the augmented samples are sufficiently different from the ground-truth ones, which further reinforces this aspect.
> - **There is no data leakage in labels**. We adopt **the same fold division** in both cross-augmentation and cross-validation. Therefore, for the one fold as the test set in cross-validation, its ground-truth labels have never been accessed either during augmentation, pre-training, or fine-tuning. As a result, the ground-truth label information of the test set is not leaked into the training processes of both Prompt-DDG (teacher) and Light-DDG (student).
>
> **Point-to-point responses.**
>
> Even though separate models are trained for each fold, they share the same architecture and objectives, leading to the learning of similar patterns regarding mutation effects.
> - The same model is capable of learning diverse mutation patterns from distinct training data, and these patterns are not necessarily similar. We divide the dataset into several folds not randomly but according to the complex structure. Hence, the complex structures and mutation patterns within that fold to be augmented may be dissimilar to the training folds. Secondly, there exists a huge gap between the random mutation patterns and the ground-truth mutation patterns that adhere to the natural distribution.
>
> The augmented data inherently carry the models' biases about mutation effects, allowing information from the test set to inadvertently influence the training process.
> - Both augmentation and distillation are anticipated to enable our model to acquire the inductive bias of the preceding SOTA method (Prompt-DDG) without any occurrence of data leakage. Since the mutations and labels of the test set have not been leaked during both augmentation and distillation, they have no direct influence on the training process.
>
> Comparing your approach to large-scale augmentation methods like AlphaFold 3 is not reasonable.
> - Thank you for your comment. We have revised (removed) the inappropriate expression in this part of the paper.
>
> **Additional experiment: what happens if a data leak occurs?**
>
> The results in Figure 5(b) demonstrate the model sensitivity to the sizes of augmentation data. On top of it, we further explore what happens if we adopt **an augmentation strategy with mutation leakage**. To simulate it, we predict the ground-truth mutations within the augmented fold and directly use them as augmentation data. The results are shown in the table below, where mutation leakage is specifically noted and the others are no leakage.
>
> | Aug. Data Size ($\times 10^4$) | 0 | 0.71 (no leakage) | 1 | 2 | 4 | 8 | 16 | 32 | 64 | 0.71 (mutation leakage) |
> | - | - | - | - | - | - | - | - | - | - | - |
> | | 0.481 | 0.485 | 0.488 | 0.510 | 0.525 | 0.539 | 0.544 | 0.548 | 0.554 | 0.527 |
>
> **Analysis:** With the same amount of augmentation data (7100~), the model with mutation leakage performs much better than the one without leakage (0.527 vs. 0.485). Our model, even without mutation leakage, benefits from a large amount of augmented data and even outperforms the model with mutation leakage (0.527 vs. 0.554). This is attributable to the fact that although our model does not have access to the real-ground mutations, it has “seen” an adequate number of mutation combinations from a large quantity of randomly generated augmentation data. The pre-training using this augmentation data contributes to enhancing the generalization ability of the model with respect to unseen mutations.
>
> We hope we have addressed your concerns in data leakage. If you need any clarification, please feel free to contact us.

---

> ### Author Response · Authors · 2024-12-02
>
> Dear Reviewer Ceec,
>
> For the concerns you left about data leakage, we have provided further clarification and additional experiments. We hope you take the time to have a look. In our opinion, there is no data leakage here. The key operation for this is that we have used random mutations instead of direct ground-truth mutations in the test set during augmentation, enabling the **random mutation patterns** in the augmented data to be different from the **natural mutation patterns** in the training data. Further, we provide an additional experiment (what happens if there is a data leakage?) to illustrate this point. We observe that the performance benefits of our method do not come directly from data (mutation) leakage, but rather from a larger ratio of augmentation data. This enables the model to see **more but random** combinations of mutations, enhancing generalization to unknown mutations.
>
> Considering that the deadline (December 2, AoE) for author-reviewer discussions is approaching, we would like to confirm that your concerns above have been addressed.
>
> If you still have questions about this, please do not hesitate to contact us.
>
> Best regards,
>
> Authors.

---

### Official Review · Reviewer_6abJ · 2024-11-03

**Soundness:** 3
**Presentation:** 3
**Contribution:** 3
**Rating:** 8
**Confidence:** 3

**Summary:**

In this study, a transformer-based model utilizing protein structural information is proposed to predict the change in binding free energy of protein complexes (antibodies). The simpler architecture of their model as compared to the previous SOTA, allows for faster exploration of fitness landscape in the antibody design task. Their model trained with data augmentation, and teacher guidance by the previous SOTA (prompt-DDG) has been shown to have better performance in the prediction of change in binding free energy.

Additionally, a novel Mutation Explainer algorithm, utilizing the predictions of their transformer-based model, is proposed to score the marginal benefit of each mutation per site. In an application to SARS CoV-2, their algorithm is shown to be better than generative or energy-based models in ranking known effective mutants, and better than conditional generative models in optimizing antibody mutants against SARS CoV-2.

Their evaluations are accompanied by comprehensive ablation study on the choice of teacher model, inclusion of data augmentation, knowledge distillation and structural information in the training of predictive model.

**Strengths:**

1. Generated an augmented SKEMPI-v2 dataset using previous SOTA model predictions on random mutations performed on SKEMPI-v2 annotated dataset. The augmented dataset was used for pretraining of their model.

2. Demonstrating the utility of an effective predictor of delta binding free energy in antibody optimization and screening, i.e., mutant selection, without the need for generative modeling.

3. Extensive benchmark on delta binding free energy prediction and ablation study on various design choices including type of teacher model, data augmentation, knowledge distillation and protein structural information.

**Weaknesses:**

1. Limited performance evaluation on anti-body design/optimization tasks.

2. A discussion on how their prior based design can be combine with other techniques, such as generative modeling to improve antibody design

3. Brief explanation as to why the bar is low (based on the range of correlation values in the tables) in the prediction of delta binding free energy.

**Questions:**

1. In the experiments of Table 6, the generated antibody mutants with different strategies are evaluated by the delta binding free energy predictor used to design the mutation explainer. This may not be a fair comparison, as it is kind of expected for the mutation explainer to perform well evaluated by the same predictor as the one that it is built upon.
I understand that the ground truths are not available, however the performance evaluation should be independent of the model design.

2. I am curious to know if you have examined the transfer from single to multiple site mutations in the prediction of delta binding free energy? if yes, how do the current models perform? This is a question of interest in protein property prediction tasks using mutagenesis datasets.

---

> ### Author Response · Authors · 2024-11-18
> **Response to Reviewer 6abJ - Part (1/2)**
>
> Thanks for your insightful comments! We have marked the revised or newly added contents in red in the revised manuscript, and address the concerns you raised one by one as below:
>
> ---
>
> **Q1**: Limited performance evaluation on anti-body design/optimization tasks.
>
> **A1**: Existing experiments on antibody design that we have provided in the original manuscript:
> - (1) antibody screening and ranking (**Table 5**);
> - (2) a case study of antibody optimization (**Table 6**);
> - (3) antibody optimization on SAbDab (**Table A1**);
> - (4) mutation explanations (**Figure 6**).
>
> Additional thee experiments we will add in the revised manuscript.
>
> - (1) Independent metrics for performance evaluation (see our response **A4** to **Q4**).
> - (2) Taking Light-DDG as the fitness function, we have further compared two additional search strategies, gg-dWJS and CMA-ES-based, in **Table 6**, with relevant analysis in **Lines 474-482**.
> - (3) We provide a performance comparison of random search and (preference-based) directed search under different numbers of mutation candidates. The results are shown below. It can be observed that random search nearly fails with limited search resources, whereas preference-based search performs quite well, which demonstrates its advantage of searching for promising mutations with limited resources. Besides, random search can benefit from more searches, e.g., ΔΔG from -1.865 to -2.316 when the mutation candidates are increased from 10,000 to 10,0000. However, it should be noted that the prerequisite that random search works well in the huge mutation space is the fast screening enabled by Light-DDG of this paper.
>
> | #Candidate | Method | CDR-H1 | CDR-H2 | CDR-H3 | CDR-H1/2/3 |
> | --- | ---| --- | --- | --- | --- |
> | 100        | Random | -0.173 | -0.579 | -0.042 | -0.215 |
> | 100        | Directed | -0.864 | -1.631 | -0.653 | -2.081 |
> | 1,000     | Random | -0.643 | -1.422 | -0.195 | -0.832 |
> | 1,000     | Directed | -1.179 | -2.036 | -0.818 | -2.419 |
> | 10,000   | Random | -1.063 | -1.865 | -0.534 | -1.325 |
> | 10,000   | Directed | -1.241 | -2.192 | -0.946 | -2.872 |
> | 100,000 | Random | -1.320 | -2.316 | -0.730 | -1.418 |
> | 100,000 | Directed | -1.418 | -2.408 | -1.106 | -3.151 |
>
> ---
>
> **Q2**: A discussion on how their prior based design can be combine with other techniques, such as generative modeling to improve antibody design.
>
> **A2**: Thanks for your constructive suggestion. We have added a discussion of how to combine the prior-based design of this paper with deep generative models in **Lines 536-539**. We believe that constructing preference pairs using Light-DDG for preference alignment and using Light-DDG as guidance in diffusion models for controllable generation are two promising solutions.
>
> ---
>
> **Q3**: Brief explanation as to why the bar is low (based on the range of correlation values in the tables) in the prediction of delta binding free energy.
>
> **A3**: We can indeed observe from tables, such as **Table 2**, that the metrics of those sequence-based methods are much lower than others (perhaps this is what you mean by “bar is low”?). This is because these sequence-based methods were initially developed to predict mutational effects for single proteins, such as the changes in the stability, fluorescence, fitness, etc., which makes it hard to be directly extended for protein-protein interactions. It is well known that protein-protein interactions are largely determined by protein structure rather than sequence. Hence, predicting the effect of mutations on protein complexes requires more efficient use of protein structures, causing those purely sequence-based methods to perform poorly.

---

> ### Author Response · Authors · 2024-11-18
> **Response to Reviewer 6abJ - Part (2/2)**
>
> **Q4**: I understand that the ground truths are not available, however the performance evaluation should be independent of the model design.
>
> **A4**: One bottleneck in evaluating the quality of generated antibodies is that the ground truth is often not available, as you said. Thus, previous practices [1,2,3] have usually resorted to a well-performing ΔΔG predictor for computational evaluation. From the numerical metrics in **Table 2**, our proposed Light-DDG is the best performer among all methods.
>
> However, we agree with you that “the performance evaluation should be independent of the model design”. To fulfill this, we replace Light-DDG with the average results of four independent ΔΔG predictors (FoldX, RDE-Network, MIF-Network, DiffAffinity) and re-report **Table 6** below.
>
> | Method | CDR-H1 | CDR-H2 | CDR-H3 | CDR-H1/2/3 |
> | ---| --- | --- | --- | --- |
> | RefineGNN | -0.490 | -1.379 | -0.106 | - |
> | MEAN         | -0.655 | -1.751 | -0.669 | - |
> | DiffAb          | -1.083 | -1.832 | -0.875 | - |
> | dyMEAN     | -0.890 | -1.951 | -0.756 | - |
> | Random      | -1.103 | -1.765 | -0.494 | -1.290 |
> | Directed      | **-1.328** | **-2.184** | **-1.048** | **-2.924** |
>
> Same observations can be derived: (1) Uni-Anti leads to lower ΔΔG mutants regardless of which CDR is optimized; (2) joint optimization of three CDRs leads to larger performance gains.
>
> [1] Kong, et. al. Conditional antibody design as 3d equivariant graph translation. ICLR 2023.
>
> [2] Kong, et. al. End-to-end full-atom antibody design. ICML 2023.
>
> [3] Tan, et. al. Cross-Gate MLP with Protein Complex Invariant Embedding is A One-Shot Antibody Designer. AAAI 2024.
>
> ---
>
> **Q5**: I am curious to know if you have examined the transfer from single to multiple site mutations in the prediction of delta binding free energy? if yes, how do the current models perform?
>
> **A5**: We have provided performance comparisons of the various methods under single and multiple site mutations in **Table 3** and the corresponding analyses in **Lines 374-377**. The results show that two state-of-the-art methods, Prompt-DDG and ProMIM, each have strengths in different metrics and mutation settings. However, our Light-DDG greatly outperforms all baselines for both single- and multiple-site mutations, especially the more challenging multi-site mutations.
>
> ---
>
> In light of these responses, we hope we have addressed your concerns. If we have left any notable points of concern unaddressed, please do share and we will attend to these points. We sincerely hope that you can appreciate our efforts on responses and revisions and thus raise your score.

---

> > ### Comment · Reviewer_6abJ · 2024-11-25
> > **Score is raised to 8**
> >
> > Thanks for answering my questions. This is a very nice paper. I raise my score to 8.

---

> > > ### Author Response · Authors · 2024-12-02
> > >
> > > Many thanks for appreciating our response and raising your score to 8. It is encouraging to hear from you that “This is a very nice paper”.

---

> ### Author Response · Authors · 2024-11-25
>
> Dear Reviewer 6abJ,
>
> Thank you for your insightful and helpful comments once again. We have thoroughly considered all the feedback and provided a point-to-point response to you.
>
> The deadline for author-reviewer discussions is approaching. If you still need any clarification or have any other questions, please do not hesitate to let us know. We hope that our response addresses your concerns to your satisfaction and we would like to kindly request a reconsideration of a higher rating if it is possible.
>
> Best regards,
>
> Authors.

---

### Meta-Review · Area_Chair_PbKS · 2024-12-20

**Metareview:**

This paper constructs a lightweight supervised predictor of binding free energy ($\Delta\Delta G$) by constructing an augmented dataset of binding affinities and then training against a combination of this dataset's labels and knowledge distillation from Prompt-DDG. The authors use their model to explain the value of mutations at individual sites in the antibody sequence via Shapley values, and using their model as an oracle to search for DDG improving mutations via sampling. The empirical results are strong, with the authors' methods outperforming several larger recent models for these tasks.

**Additional Comments On Reviewer Discussion:**

Several reviewers noted a few minor weaknesses, such as a relatively small number of results on antibody design and optimization problems, with the SARS-CoV 2 benchmark being the primary study here. Although the majority of reviewers didn't engage the authors in the discussion period, the overall ratings were quite strong going into the discussion period and there was nearly unanimous support for even the original version of the paper. Despite this, the authors have clearly put in the effort to address these concerns with new results and updates to the original submission PDF.

---

### Decision · Program_Chairs · 2025-01-22

Accept (Poster)